# WiSE-ALE: Wide Sample Estimator for Aggregate Latent Embedding

**Shuyu Lin, Stephen Roberts & Niki Trigoni**
University of Oxford
Oxford, UK
`{slin, sjrob}@robots.ox.ac.uk`
`niki.trigoni@cs.ox.ac.uk`

**Ronald Clark**
Imperial College London
London, UK
`ron.clark@live.com`

**Robert Birke**
ABB Corporate Research
Switzerland
`robert.birke@ch.abb.com`

## Abstract

In this paper, we present a new generative model for learning latent embeddings. Compared to the classical generative process, where each observed data point is generated from an individual latent variable, our approach assumes *a global latent variable* to generate the whole set of observed data points. We then propose a learning objective that is derived as an approximation to a lower bound to the data log likelihood, leading to our algorithm, WiSE-ALE. Compared to the standard ELBO objective, where the variational posterior for each data point is encouraged to match the prior distribution, the WiSE-ALE objective matches the averaged posterior, over all samples, with the prior, allowing the sample-wise posterior distributions to have a wider range of acceptable embedding mean and variance and leading to better reconstruction quality in the auto-encoding process. Through various examples and comparison to other state-of-the-art VAE models, we demonstrate that WiSE-ALE has excellent information embedding properties, whilst still retaining the ability to learn a smooth, compact representation.

## 1 Introduction

Unsupervised learning is a central task in machine learning. Its objective can be informally described as learning a representation of some observed forms of information in a way that the representation summarizes the overall statistical regularities of the data (Barlow, 1989). Deep generative models are a popular choice for unsupervised learning, as they marry deep learning with probabilistic models to estimate a joint probability between high dimensional input variables $\mathbf{x}$ and unobserved latent variables $\mathbf{z}$. Early successes of deep generative models came from Restricted Boltzmann Machines (Hinton & Salakhutdinov, 2006) and Deep Boltzmann Machines (Salakhutdinov & Hinton, 2009), which aim to learn a compact representation of data. However, the fully stochastic nature of the network requires layer-by-layer pre-training using MCMC-based sampling algorithms, resulting in heavy computation cost.

Kingma & Welling (2013) consider the objective of optimizing the parameters in an auto-encoder network by deriving an analytic solution to a variational lower bound of the log likelihood of the data, leading to the Auto-Encoding Variational Bayes (AEVB) algorithm. They apply a reparameterization trick to maximally utilize deterministic mappings in the network, significantly simplifying the training procedure and reducing instability. Furthermore, a regularization term naturally occurs in their model, allowing a prior $p(\mathbf{z})$ to be placed over every sample embedding $q(\mathbf{z}|\mathbf{x})$. As a result, the learned representation becomes compact and smooth; see e.g. Fig. 1 where we learn a 2D embedding of MNIST digits using 4 different methods and visualize the aggregate posterior distribution of 64 random samples in the learnt 2D embedding space.

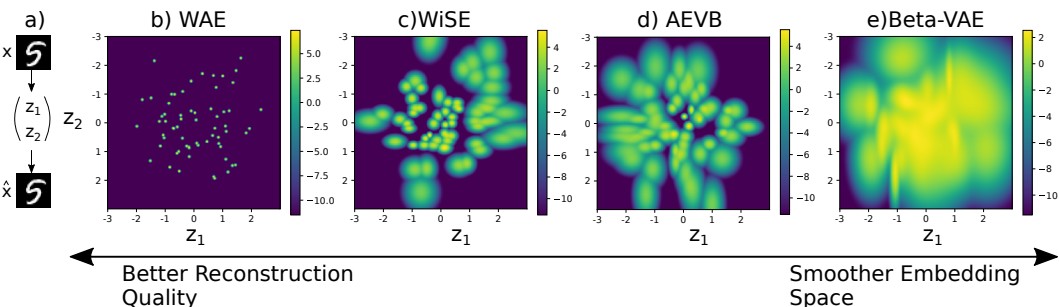

Figure 1: (a) Learning a 2D embedding of MNIST handwritten digits through an auto-encoding framework. Embedding distributions (aggregate posteriors) of 64 randomly drawn MNIST digits when WAE (b), our proposed WiSE-ALE (c), AEVB (d) or $\beta$-VAE (e) is used for the learning. Different learning algorithms find a different level of tradeoff between the reconstruction quality (information preservation) and the smoothness of the posterior distribution.

However, because the choice of the prior is often uninformative, the smoothness constraint imposed by this regularization term can cause information loss between the input samples and the latent embeddings, as shown by the merging of individual embedding distributions in Fig. 1(d) (especially in the outer areas away from zero code). Extreme effects of such behaviours can be noticed from $\beta$-VAE (Higgins et al., 2016), a derivative algorithm of AEVB which further increases the weighting on the regularizing term with the aim of learning an even smoother, disentangled representation of the data. As shown in Fig. 1(e), the individual embedding distributions are almost indistinguishable, leading to an overly severe information bottleneck which can cause high rates of distortion (Tishby et al., 1999). In contrast, perfect reconstruction can be achieved using WAE (Tolstikhin et al., 2017), but the learnt embedding distributions appear to severely non-smooth (Fig. 1(b)), indicating a small amount of noise in the latent space would cause generation process to fail.

In this paper, we propose WiSE-ALE (a wide sample estimator), which imposes a prior on the *bulk statistics* of a mini-batch of latent embeddings. Learning under our WiSE-ALE objective does not penalize individual embeddings lying away from the zero code, so long as the aggregate distribution (the average of all individual embedding distributions) does not violate the prior significantly. Hence, our approach mitigates the distortion caused by the current form of the prior constraint in the AEVB objective. Furthermore, the objective of our WiSE-ALE algorithm is derived by applying variational inference in a simple latent variable model (Section 2) and with further approximation, we derive an analytic form of the learning objective, resulting in efficient learning algorithm.

In general, the latent representation learned using our algorithm enjoys the following properties: 1) **smoothness**, as indicated in Fig. 1(d), the probability density for each individual embedding distribution decays smoothly from the peak value; 2) **compactness**, as individual embeddings tend to occupy a maximal local area in the latent space with minimal gaps in between; and 3) **separation**, indicated by the narrow, but clear borders between neighbouring embedding distributions as opposed to the merging seen in AEVB. In summary, our contributions are:

- proposing a new latent variable model that uses **a single global latent variable** to generate the entire dataset,

- deriving **a variational lower bound** to the data log likelihood in our latent variable model, which allows us to impose prior constraint on the *bulk statistics* of a mini-batch embedding distributions,

- and deriving **analytic approximations** to the lower bound, leading to our efficient WiSE-ALE learning algorithm.

In the rest of the paper, we first review directed graphical models in Section 2. We then derive our variational lower bound and its analytic approximations in Section 3. Related work is discussed in Section 4. Experiment results are analyzed in Section 5, leading to conclusions in Section 6.

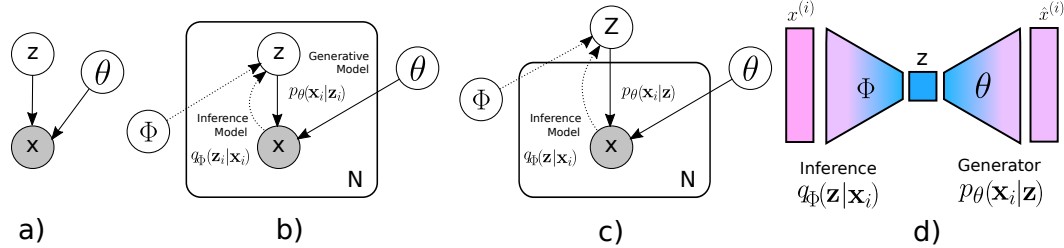

Figure 2: (a) Directed graphical models for the generative model between observation $\mathbf{x}$ and latent variable $\mathbf{z}$. (b) Latent variable model for the AEVB algorithm, where $z$ indicates $N$ random variables for $N$ latent codes. (c) Latent variable model for the our WiSE-ALE algorithm, where $z$ is a single random variable for the aggregate posterior of the entire dataset. (d) A generic neural network implementation suitable for both (b) and (c).

## 2 BACKGROUND: LATENT VARIABLE MODELS

Here we introduce the latent variable model used in our WiSE-ALE algorithm and compare with the latent variable model used in the AEVB algorithm (Kingma & Welling, 2013).

Given $N$ observations of input samples $\boldsymbol{x} \in R^{d_x}$ denoted $\mathcal{D}_N = \left( \boldsymbol{x}^{(1)}, \boldsymbol{x}^{(2)}, \cdots, \boldsymbol{x}^{(N)} \right)$, we assume $\mathbf{x}$ is generated from a latent variable $\mathbf{z} \in R^{d_z}$ of a much lower dimension. Here we denote $\mathbf{x}$ and $\mathbf{z}$ as random variables, $\boldsymbol{x}^{(i)}$ or $\boldsymbol{z}^{(i)}$ as the $i$-th input or latent code sample (i.e. a vector), and $\mathbf{x}_i$ and $\mathbf{z}_i$ as the random variable for $\boldsymbol{x}^{(i)}$ and $\boldsymbol{z}^{(i)}$. As shown in Fig. 2(a), this generative process can be modelled by a simple directed graphical model (Jordan et al., 1999), which models the joint probability distribution $p_\theta(\mathbf{x}, \mathbf{z}|\mathcal{D}_N) = p_\theta(\mathbf{x}|\mathbf{z})p(\mathbf{z}|\mathcal{D}_N) = p_\theta(\mathbf{z}|\mathbf{x})p(\mathbf{x}|\mathcal{D}_N)$ between $\mathbf{x}$ and $\mathbf{z}$, given the current observations $\mathcal{D}_N$. $p(\mathbf{z}|\mathcal{D}_N)$ is the latent distribution given $\mathcal{D}_N$, $p(\mathbf{x}|\mathcal{D}_N)$ is the data distribution for $\mathcal{D}_N$ and $p_\theta(\mathbf{x}|\mathbf{z})$ and $p_\theta(\mathbf{z}|\mathbf{x})$ denote the complex transformation from the latent to the input space and reverse, where the transformation mapping is parameterised by $\theta$. The learning task is to estimate the optimal set of $\boldsymbol{\theta}$ so that this latent variable model can explain the data $\mathcal{D}_N$ well.

As the inference of the latent variable $\mathbf{z}$ given $\mathbf{x}$ (i.e. $p_\theta(\mathbf{z}|\mathbf{x})$) cannot be directly estimated because $p(\mathbf{x}|\mathcal{D}_N)$ is unknown, both AEVB (Fig. 2(b)) and our WiSE-ALE (Fig. 2(c)) resort to variational method to approximate the target distribution $p_\theta(\mathbf{z}|\mathbf{x})$ by a proposal distribution $q_\phi(\mathbf{z}|\mathbf{x})$ with the modified learning objective that both $\theta$ and $\phi$ are optimised so that the model can explain the data well and $q_\phi(\mathbf{z}|\mathbf{x})$ approaches $p_\theta(\mathbf{z}|\mathbf{x})$. The primary difference between the AEVB model and our WiSE-ALE model lies in how the joint probability $p_\theta(\mathbf{x}, \mathbf{z}|\mathcal{D}_N)$ is modelled and specifically whether we assume an individual random variable for each latent code $\boldsymbol{z}^{(i)}$. The AEVB model assumes a pair of random variables $(\mathbf{x}_i, \mathbf{z}_i)$ for each $\boldsymbol{x}^{(i)}$ and estimates the joint probability as

$$p_\theta(\mathbf{x}, \mathbf{z}|\mathcal{D}_N) = p_\theta(\mathbf{x}_1, \mathbf{x}_2, \cdots, \mathbf{x}_N, \mathbf{z}_1, \mathbf{z}_2, \cdots, \mathbf{z}_N | \mathcal{D}_N) \tag{1}$$

$$= p_\theta(\mathbf{x}_1, \mathbf{x}_2, \cdots, \mathbf{x}_N | \mathbf{z}_1, \mathbf{z}_2, \cdots, \mathbf{z}_N) \, p_\theta(\mathbf{z}_1, \mathbf{z}_2, \cdots, \mathbf{z}_N | \mathcal{D}_N) \tag{2}$$

$$= \prod_{i=1}^{N} p_\theta(\mathbf{x}_i | \mathbf{z}_1, \mathbf{z}_2, \cdots, \mathbf{z}_N) \prod_{i=1}^{N} p_\theta(\mathbf{z}_i | \mathcal{D}_N) \tag{3}$$

$$= \prod_{i=1}^{N} p_\theta(\mathbf{x}_i | \mathbf{z}_i) \prod_{i=1}^{N} p_\theta(\mathbf{z}_i | \mathcal{D}_N) \tag{4}$$

$$= \prod_{i=1}^{N} \left( p_\theta(\mathbf{x}_i | \mathbf{z}_i) \, p_\theta(\mathbf{z}_i | \mathcal{D}_N) \right). \tag{5}$$

The equality between Eq. 2 and Eq. 3 can only be made with the assumption that the generation process for each $\mathbf{x}_i$ is independent (first product in Eq. 3) and each $\mathbf{z}_i$ is also independent (second product in Eq. 3). Such interpretation of the joint probability leads to the latent variable model in Fig. 2(b) and the prior constraint (often taken as $\mathcal{N}(0, \boldsymbol{I})$ to encourage shrinkage when no data is observed) is imposed on every $\mathbf{z}_i$.

In contrast, our WiSE-ALE model takes a single random variable to estimate the latent distribution for the entire dataset $\mathcal{D}_N$. Hence, the joint probability in our model can be broken down as

$$p_\theta(\mathbf{x}, \mathbf{z} | \mathcal{D}_N) = p_\theta(\mathbf{x}_1, \mathbf{x}_2, \cdots, \mathbf{x}_N, \mathbf{z} | \mathcal{D}_N) \tag{6}$$

$$= p_\theta(\mathbf{x}_1, \mathbf{x}_2, \cdots, \mathbf{x}_N, | \mathbf{z}) \, p_\theta(\mathbf{z} | \mathcal{D}_N) \tag{7}$$

$$= p_\theta(\mathbf{z} | \mathcal{D}_N) \prod_{i=1}^{N} p_\theta(\mathbf{x}_i | \mathbf{z}), \tag{8}$$

leading to the latent variable model illustrated in Fig. 2(c). The only assumption we make in our model is assuming the generative process of different input samples given the latent distribution of the current dataset as independent, which we consider as a sensible assumption. More significantly, we do not require independence between different $\mathbf{z}_i$ as opposed to the AEVB model, leading to a more flexible model. Furthermore, the prior constraint in our model is naturally imposed on the aggregate posterior $p(\mathbf{z}|\mathcal{D}_N)$ for the entire dataset, leading to more flexibility for each individual sample latent code to shape an embedding distribution to preserve a better quality of information about the corresponding input sample.

Neural networks can be used to parameterize $p_\theta(\mathbf{x}_i|\mathbf{z}_i)$ in the generative model and $q_\phi(\mathbf{z}_i|\mathbf{x}_i)$ in the inference model from the AEVB latent variable model or $p_\theta(\mathbf{x}_i|\mathbf{z})$ and $q_\phi(\mathbf{z}|\mathbf{x}_i)$ correspondingly from our WiSE-ALE latent variable model. Both networks can be implemented through an auto-encoder network illustrated in Fig. 2(d).

## 3 OUR METHOD

In this section, we first define the aggregate posterior distribution $p(\mathbf{z}|\mathcal{D}_N)$ which serves as a core concept in our WiSE-ALE proposal. We then derive a variational lower bound to the data log likelihood $\log p(\mathcal{D}_N)$ using $p(\mathbf{z}|\mathcal{D}_N)$. Further, analytic approximation to the lower bound is derived, allowing efficient optimization of the model parameters and leading to our WiSE-ALE learning algorithm. Intuition of our proposal is also discussed.

### 3.1 AGGREGATE POSTERIOR

Here we formally define the aggregate posterior distribution $p(\mathbf{z}|\mathcal{D}_N)$, i.e. the latent distribution given the entire dataset $\mathcal{D}_N$. Considering

$$p(\mathbf{z}|\mathcal{D}_N) = \int p_\theta(\mathbf{z}|\mathbf{x}) p(\mathbf{x}|\mathcal{D}_N) \mathrm{d}\mathbf{x} = \sum_{i=1}^{N} p_\theta(\mathbf{z}|\mathbf{x} = \boldsymbol{x}^{(i)}) P(\mathbf{x} = \boldsymbol{x}^{(i)}|\mathcal{D}_N) = \frac{1}{N} \sum_{i=1}^{N} p_\theta(\mathbf{z}|\boldsymbol{x}^{(i)}), \tag{9}$$

we have the aggregate posterior distribution for the entire dataset as the average of all the individual sample posteriors. The second equality in Eq. 9 is made by approximating the integral through summation. The third equality is obtained following the conventional assumption in the VAE literature that each input sample, $\boldsymbol{x}^{(i)}$, is drawn from the dataset $\mathcal{D}_N$ with equal probability, i.e. $P(\boldsymbol{x}^{(i)}|\mathcal{D}_N) = \frac{1}{N}$. Similarly, for the estimated aggregate posterior distribution $q(\mathbf{z}|\mathcal{D}_N)$, we have

$$q(\mathbf{z}|\mathcal{D}_N) = \frac{1}{N} \sum_{i=1}^{N} q_\phi(\mathbf{z}|\boldsymbol{x}^{(i)}). \tag{10}$$

### 3.2 ALTERNATIVE VARIATIONAL LOWER BOUND (LB)

To carry out variational inference, we minimize the KL divergence between the estimated and the true aggregate posterior distributions $q_\phi(\mathbf{z}|\mathcal{D}_N)$ and $p_\theta(\mathbf{z}|\mathcal{D}_N)$, i.e.

$$D_{\mathrm{KL}}\big[\, q_\phi(\mathbf{z}|\mathcal{D}_N) \| p_\theta(\mathbf{z}|\mathcal{D}_N)\big] \;=\; \mathbb{E}_{q_\phi(\mathbf{z}|\mathcal{D}_N)}\left[\log \frac{q_\phi(\mathbf{z}|\mathcal{D}_N)}{p_\theta(\mathbf{z}|\mathcal{D}_N)}\right]. \tag{11}$$

Substituting $p_\theta(\mathbf{z}|\mathcal{D}_N) = \frac{p_\theta(\mathcal{D}_N|\mathbf{z})\,p(\mathbf{z})}{p_\theta(\mathcal{D}_N)}$ in Eq. 11 and breaking down the products and fractions inside the log, we have

$$D_{\mathrm{KL}}\big[\,q_\phi(\mathbf{z}|\mathcal{D}_N)\|p_\theta(\mathbf{z}|\mathcal{D}_N)\big] = \mathbb{E}_{q_\phi(\mathbf{z}|\mathcal{D}_N)}\big[\log q_\phi(\mathbf{z}|\mathcal{D}_N) - \log p_\theta(\mathcal{D}_N|\mathbf{z}) - \log p(\mathbf{z})\big] + \log p(\mathcal{D}_N).$$

Re-arranging the above equation, we have

$$\log p(\mathcal{D}_N) - D_{\mathrm{KL}}\big[\,q_\phi(\mathbf{z}|\mathcal{D}_N)\|p_\phi(\mathbf{z}|\mathcal{D}_N)\big] = \mathbb{E}_{q_\phi(\mathbf{z}|\mathcal{D}_N)}\big[\log p_\theta(\mathcal{D}_N|\mathbf{z})\big] - D_{\mathrm{KL}}\big[\,q_\phi(\mathbf{z}|\mathcal{D}_N)\|p(\mathbf{z})\big].$$

As $D_{\mathrm{KL}}\big[q_\phi(\mathbf{z}|\mathcal{D}_N)\|p_\phi(\mathbf{z}|\mathcal{D}_N)\big]$ is non-negative, we have obtained a variational lower bound $L^{\mathrm{WiSE\text{-}ALE}}(\phi,\theta;\mathcal{D}_N)$ to the marginal log likelihood of the data $\log p(\mathcal{D}_N)$ as

$$\log p(\mathcal{D}_N) \geq L^{\mathrm{WiSE\text{-}ALE}}(\phi,\theta;\mathcal{D}_N) = \underbrace{\mathbb{E}_{q_\phi(\mathbf{z}|\mathcal{D}_N)}\big[\log p_\theta(\mathcal{D}_N|\mathbf{z})\big]}_{\text{①\ Reconstruction likelihood}} - \underbrace{D_{\mathrm{KL}}\big[\,q_\phi(\mathbf{z}|\mathcal{D}_N)\|p(\mathbf{z})\big]}_{\text{②\ Prior constraint}}.$$

$$\text{(12)}$$

There are two terms in the derived lower bound: ① a **reconstruction likelihood** term that indicates how likely the current dataset $\mathcal{D}_N$ are generated by the aggregate latent posterior distribution $q_\phi(\mathbf{z}|\mathcal{D}_N)$ and ② a **prior constraint** that penalizes severe deviation of the aggregate latent posterior distribution $q_\phi(\mathbf{z}|\mathcal{D}_N)$ from the preferred prior $p(\mathbf{z})$, acting naturally as a regularizer. By maximizing the lower bound $L^{\mathrm{WiSE\text{-}ALE}}(\phi,\theta;\mathcal{D}_N)$ defined in Eq. 12, we are approaching to $\log p(\mathcal{D}_N)$ and, hence, obtaining a set of parameters $\theta$ and $\phi$ that find a natural balance between a good reconstruction likelihood (good explanation of the observed data) and a reasonable level of compliance to the prior assumption (achieving some preferable properties of the posterior distribution, such as smoothness and compactness).

### 3.3 Approximation of the Proposed Lower Bound

To allow fast and efficient optimization of the model parameters $\theta$ and $\phi$, we derive analytic approximations for the two terms in our proposed lower bound (Eq. 12).

#### 3.3.1 Approximation to Reconstruction Likelihood Term

To approximate ① **reconstruction likelihood** term in Eq. 12, we first substitute the definition of the approximate aggregate posterior given in Eq. 10 in the expectation operation in $\mathbb{E}_{q_\phi(\mathbf{z}|\mathcal{D}_N)}\big[\log p_\theta(\mathcal{D}_N|\mathbf{z})\big]$, i.e.

$$\mathbb{E}_{q_\phi(\mathbf{z}|\mathcal{D}_N)}\big[\log p_\theta(\mathcal{D}_N|\mathbf{z})\big] = \frac{1}{N}\sum_{i=1}^{N} \mathbb{E}_{q_\phi(\mathbf{z}|\boldsymbol{x}^{(i)})}\big[\log p_\theta(\mathcal{D}_N|\mathbf{z})\big]. \tag{13}$$

Now we can decompose the $p_\theta(\mathcal{D}_N|\mathbf{z})$ as a product of individual sample likelihood, due to the conditional independence, i.e.

$$\log p_\theta(\mathcal{D}_N|\mathbf{z}) = \log \prod_{j=1}^{N} p_\theta(\boldsymbol{x}^{(j)}|\mathbf{z}) = \sum_{j=1}^{N} \log p_\theta(\boldsymbol{x}^{(j)}|\mathbf{z}). \tag{14}$$

Substituting this into Eq. 13, we have

$$\mathbb{E}_{q_\phi(\mathbf{z}|\mathcal{D}_N)}[\log p_\theta(\mathcal{D}_N|\mathbf{z})] = \sum_{i=1}^{N} \mathbb{E}_{q_\phi(\mathbf{z}|\boldsymbol{x}^{(i)})}\left[\frac{1}{N}\sum_{j=1}^{N}\log p_\theta(\boldsymbol{x}^{(j)}|\mathbf{z})\right]. \tag{15}$$

Eq. 15 can be used to evaluate the reconstruction likelihood for $\mathcal{D}_N$. However, learning directly with this reconstruction estimate does not lead to convergence in our experiments. We choose to simplify the reconstruction likelihood further to be able to reach convergence during learning at the cost of losing the lower bound property of the objective function $L^{\mathrm{WiSE\text{-}ALE}}(\phi,\theta;\mathcal{D}_N)$. Firstly, we apply Jensen inequality to the term inside the expectation in Eq. 15, leading to an upper bound of the reconstruction likelihood term as

$$\mathbb{E}_{q_\phi(\mathbf{z}|\mathcal{D}_N)}[\log p_\theta(\mathcal{D}_N|\mathbf{z})] \leq \sum_{i=1}^{N} \mathbb{E}_{q_\phi(\mathbf{z}|\boldsymbol{x}^{(i)})}\left[\log\left(\frac{1}{N}\sum_{j=1}^{N} p_\theta(\boldsymbol{x}^{(j)}|\mathbf{z})\right)\right]. \tag{16}$$

Now $(N-1)$ sample-wise likelihood distributions in the summation inside the $\log$ can be dropped with the assumption that the $p_\theta(\boldsymbol{x}^{(j)}|\mathbf{z})$ will only be non-zero if $\mathbf{z}$ is sampled from the posterior distribution of the same sample $\boldsymbol{x}^{(j)}$ at the encoder, i.e. $i = j$. Therefore, the approximation becomes

$$\mathbb{E}_{q_\phi(\mathbf{z}|\mathcal{D}_N)}[\log p_\theta(\mathcal{D}_N|\mathbf{z})] \leq \sum_{i=1}^{N} \mathbb{E}_{q_\phi(\mathbf{z}|\boldsymbol{x}^{(i)})}\Big[\log p_\theta(\boldsymbol{x}^{(i)}|\mathbf{z})\Big] - N \log N. \tag{17}$$

Using the approximation of the reconstruction likelihood term given by Eq. 17 rather than Eq. 15, we are able to reach convergence efficiently during learning at the cost of the estimated objective no longer remaining a lower bound to $\log p(\mathcal{D}_N)$. Details of deriving the above approximation are given in Appendix A.

### 3.3.2 Approximation to Prior Constraint Term

The ②  **prior constraint** term $D_{\mathrm{KL}}\big[q_\phi(\mathbf{z}|\mathcal{D}_N)\|p(\mathbf{z})\big]$ in our objective function (Eq. 12) evaluates the KL divergence between the approximate aggregate posterior distribution $q_\phi(\mathbf{z}|\mathcal{D}_N)$ and a zero-mean, unit-variance Gaussian distribution $p(\mathbf{z})$. Here we assume that each sample-wise posterior distribution can be modelled by a factorial Gaussian distribution, i.e. $q_\phi(\mathbf{z}|\boldsymbol{x}^{(i)}) = \prod_{k=1}^{d_z} \mathcal{N}\big(\mathbf{z}_k|\mu_k(\boldsymbol{x}^{(i)}), \sigma_k^2(\boldsymbol{x}^{(i)})\big)$, where $k$ indicates the $k$-th dimension of the latent variable $\mathbf{z}$ and $\mu_k(\boldsymbol{x}^{(i)})$ and $\sigma_k^2(\boldsymbol{x}^{(i)})$ are the mean and variance of the $k$-th dimension embedding distribution for the input $\boldsymbol{x}^{(i)}$. Therefore, $D_{\mathrm{KL}}\big[q_\phi(\mathbf{z}|\mathcal{D}_N)\|p(\mathbf{z})\big]$ computes the KL divergence between a mixture of Gaussians (as Eq. 10) and $\mathcal{N}(0, \boldsymbol{I})$. There is no analytical solution for such KL divergences. Hence, we derive an analytic upper bound allowing for efficient evaluation.

Firstly, we substitute $q_\phi(\mathbf{z}|\mathcal{D}_N) = \frac{1}{N}\sum_{i=1}^{N} q_\phi(\mathbf{z}|\boldsymbol{x}^{(i)})$ (Eq. 10) to $D_{\mathrm{KL}}\big[q_\phi(\mathbf{z}|\mathcal{D}_N)\|p(\mathbf{z})\big]$, giving

$$D_{\mathrm{KL}}\big[q_\phi(\mathbf{z}|\mathcal{D}_N)\|p(\mathbf{z})\big] = \frac{1}{N}\sum_{i=1}^{N} \Big(\mathbb{E}_{q_\phi(\mathbf{z}|\boldsymbol{x}^{(i)})}\big[\log q_\phi(\mathbf{z}|\mathcal{D}_N)\big] - \mathbb{E}_{q_\phi(\mathbf{z}|\boldsymbol{x}^{(i)})}\big[\log p(\mathbf{z})\big]\Big). \tag{18}$$

Applying Jensen inequality, i.e. $\mathbb{E}_x\big[\log f(x)\big] \leq \log \mathbb{E}_x\big[f(x)\big]$, to the first term inside the summation in Eq. 18, we have

$$D_{\mathrm{KL}}\big[q_\phi(\mathbf{z}|\mathcal{D}_N)\|p(\mathbf{z})\big] \leq \mathrm{KL}_{\mathrm{approx}}^{\mathrm{UB}} \tag{19}$$

$$= \frac{1}{N}\sum_{i=1}^{N}\Big(\log \mathbb{E}_{q_\phi(\mathbf{z}|\boldsymbol{x}^{(i)})}\big[q_\phi(\mathbf{z}|\mathcal{D}_N)\big]\Big) - \frac{1}{N}\sum_{i=1}^{N}\Big(\mathbb{E}_{q_\phi(\mathbf{z}|\boldsymbol{x}^{(i)})}\big[\log p(\mathbf{z})\big]\Big). \tag{20}$$

Taking advantage of the Gaussian assumption for $q_\phi(\mathbf{z}|\boldsymbol{x}^{(i)})$ and $p(\mathbf{z})$, we can compute the expectations in Eq. 20 analytically with the result quoted below and the full derivation given in Appendix B.

$$\mathrm{KL}_{\mathrm{approx}}^{\mathrm{UB}} = \frac{1}{N}\sum_{i=1}^{N}\log\left(\frac{1}{N}\sum_{j=1}^{N}\prod_{k=1}^{d_z} A^{-1/2}B\right) + \frac{1}{2N}\sum_{i=1}^{N}\sum_{k=1}^{d_z} C, \tag{21}$$

$$\text{where } A = 2\pi\big((\sigma_k^{(i)})^2 + (\sigma_k^{(j)})^2\big), \tag{22}$$

$$B = \exp\left(-\frac{1}{2}\frac{(\mu_k^{(i)} - \mu_k^{(j)})^2}{(\sigma_k^{(i)})^2 + (\sigma_k^{(j)})^2}\right), \tag{23}$$

$$C = (\sigma_k^{(i)})^2 + (\mu_k^{(i)})^2 + \log 2\pi. \tag{24}$$

When the overall objective function $L^{\mathrm{WiSE\text{-}ALE}}(\phi, \theta; \mathcal{D}_N)$ in Eq. 12 is maximised, this upper bound approximation will approach the true KL divergence $D_{\mathrm{KL}}\big[q_\phi(\mathbf{z}|\mathcal{D}_N)\|p(\mathbf{z})\big]$, which ensures that the prior constraint on the overall aggregate posterior distribution takes effects.

### 3.3.3 Overall Objective Functions

Combining results from Section 3.3.1 and 3.3.2, we obtain an analytic approximation $L_{\mathrm{approx}}^{\mathrm{WiSE\text{-}ALE}}(\phi, \theta; \mathcal{D}_N)$ for the variational lower bound $L^{\mathrm{WiSE\text{-}ALE}}(\phi, \theta; \mathcal{D}_N)$ defined in Eq. 12, as

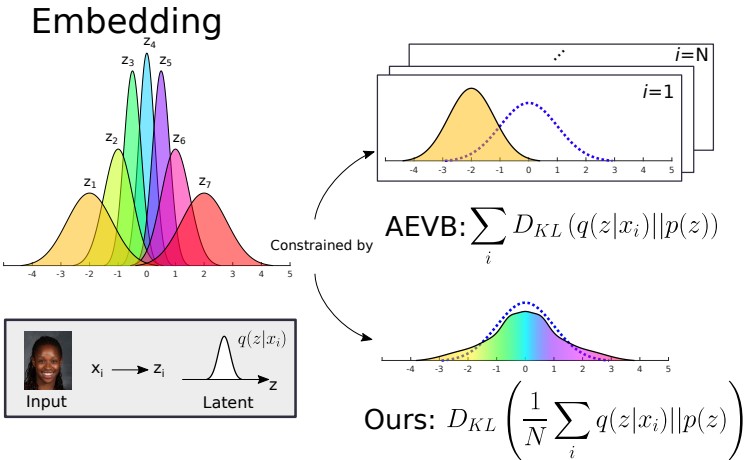

Figure 3: Comparison between our WiSE-ALE learning scheme and the AEVB estimator. AEVB imposes the prior constraint on every sample embedding distribution, whereas our WiSE-ALE imposes the constraint to the overall aggregate embedding distribution over the entire dataset (over a mini-batch as an approximation for efficient learning).

shown below:

$$L_{\text{approx}}^{\text{WiSE-ALE}}(\phi, \theta; \mathcal{D}_N) = \sum_{i=1}^{N} \mathcal{L}(\phi, \theta | \boldsymbol{x}^{(i)}) - KL\big[q_\phi(\mathbf{z}|\mathcal{D}_N) \| p(\mathbf{z})\big], \tag{25}$$

where we use $\mathcal{L}(\phi, \theta \,|\, \boldsymbol{x}^{(i)})$ to denote the sample-wise reconstruction likelihood $\mathbb{E}_{q_\phi(\mathbf{z}|\boldsymbol{x}^{(i)})}\Big[\log p_\theta(\boldsymbol{x}^{(i)}|\mathbf{z})\Big]$ given by Eq. 17 and the KL divergence term is estimated through $\text{KL}_{\text{approx}}^{\text{UB}}$ defined in Eq. 21. Optimizing $L_{\text{approx}}^{\text{WiSE-ALE}}(\phi, \theta; \mathcal{D}_N)$ w.r.t the model parameters $\phi$ and $\theta$, we are able to learn a model that naturally balances between a good embedding of the observed data and some preferred properties of the latent embedding distributions, such as smoothness and compactness.

## 3.4 COMPARISON OF AEVB AND WiSE-ALE LEARNING OBJECTIVE FUNCTIONS

Comparing the objective function in our WiSE-ALE algorithm and that proposed in AEVB algorithm (Kingma & Welling, 2013) stated below,

$$L^{\text{AEVB}}(\phi, \theta; \mathcal{D}_N) = \sum_{i=1}^{N} \mathcal{L}(\phi, \theta | \boldsymbol{x}^{(i)}) - \sum_{i=1}^{N} D_{\text{KL}}\big[q_\phi(\mathbf{z}|\boldsymbol{x}^{(i)}) \| p(\mathbf{z})\big]. \tag{26}$$

we notice that the difference lies in the form of prior constraint and the difference is illustrated in Fig. 3. AEVB learning algorithm imposes the prior constraint on every sample embedding distribution and any deviation away from the zero code or the unit variance will incur penalty. This will cause problems, as different samples cannot be simultaneously embedded to the zero code. Furthermore, when the model becomes more certain about the embedding of a specific sample as the learning continues, it will naturally favour a posterior distribution of small variance (e.g. less than 1). In contrast, our WiSE-ALE learning objective imposes the prior constraint on the aggregate posterior distribution, i.e. the average of all the sample embeddings. Such prior constraint will allow more flexibility for each sample posterior to settle at a mean and variance value in favour for good reconstruction quality, while preventing too large mean values (acting as a regulariser) or too small variance values (ensuring smoothness of the learnt latent representation).

To investigate the different behaviours of the two prior constraints more concretely, we consider only two embedding distributions $q(\mathbf{z}|\boldsymbol{x}^{(1)})$ and $q(\mathbf{z}|\boldsymbol{x}^{(2)})$ (red dashed lines) in a 1D latent space, as shown in Fig. 4. The mean values of the two embedding distributions are fixed to make the analysis simple and their variances are allowed to change. When the variances of the two embedding distributions

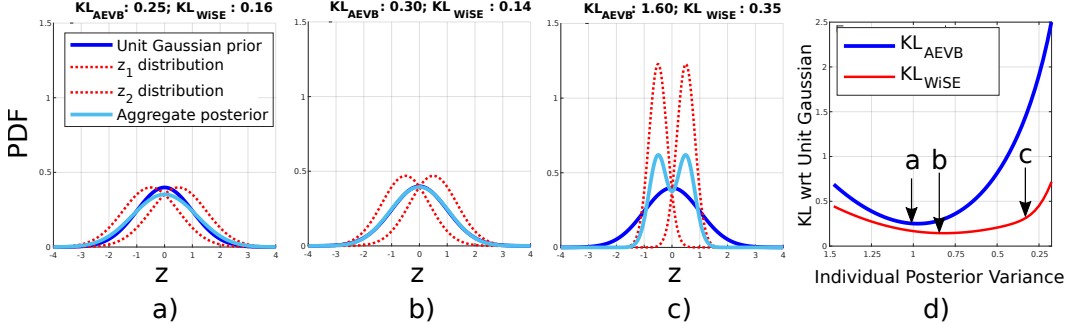

Figure 4: Compared to AEVB algorithm, our WiSE-ALE algorithm uses a prior constraint that accepts a wider range of embedding mean and embedding variance. In a-c, red dashed lines represent 2 sample-wise posterior distributions $q(\mathbf{z}|\boldsymbol{x}^{(1)})$ and $q(\mathbf{z}|\boldsymbol{x}^{(2)})$ which project the inputs $\boldsymbol{x}^{(1)}$ and $\boldsymbol{x}^{(2)}$ into the latent space (the more separable $q(\mathbf{z}|\boldsymbol{x}^{(1)})$ and $q(\mathbf{z}|\boldsymbol{x}^{(2)})$ are in the latent space, the easier it is to distinguish $\boldsymbol{x}^{(1)}$ and $\boldsymbol{x}^{(2)}$), dark blue line indicates the unit Gaussian prior distribution, light blue line shows the aggregate posterior (average of the 2 embedding distributions). (a) and (b) illustrate the most preferable embedding mean and variance obtained in AEVB algorithm and our algorithm respectively. (c) shows an acceptable mean and variance choice in our algorithm, but this choice of mean and variance would be highly penalised in AEVB. (d) Comparison of the prior constraint loss across different embedding variances in both methods.

are large, such as Fig. 4(a), $q(\mathbf{z}|\boldsymbol{x}^{(1)})$ and $q(\mathbf{z}|\boldsymbol{x}^{(2)})$ have a large area of overlap and it is difficult to distinguish the input samples $\boldsymbol{x}^{(1)}$ and $\boldsymbol{x}^{(2)}$ in the latent space. On the other hand, when the two embedding distributions have small variances, such as Fig. 4(c), there is clear separation between $\boldsymbol{x}^{(1)}$ and $\boldsymbol{x}^{(2)}$ in the latent space, indicating the embedding only introduces a small level of information loss. Overall, the prior constraint in the AEVB objective favours the embedding distributions much closer to the uninformative $\mathcal{N}(0, \boldsymbol{I})$ prior, leading to large area of overlap between the individual posteriors, whereas our WiSE-ALE objective allows a wide range of acceptable embedding mean and variance, which will then offer more flexibility in the learnt posteriors to maintain a good reconstruction quality.

### 3.5   EFFICIENT MINI-BATCH ESTIMATOR

So far our derivation has been for the entire dataset $\mathcal{D}_N$. Given a small subset $\mathcal{B}_M$ with $M$ samples randomly drawn from $\mathcal{D}_N$, we can obtain a variational lower bound for a mini-batch as:

$$L^{\text{WiSE-ALE}}(\phi, \theta; \mathcal{B}_M) = \sum_{i=1}^{M} \Big( \mathbb{E}_{q_\phi(\mathbf{z}|\boldsymbol{x}^{(i)})}\big[\log p_\theta(\boldsymbol{x}^{(i)}|\mathbf{z})\big]\Big) - D_{\text{KL}}\big[\, q_\phi(\mathbf{z}|\mathcal{B}_M)\|p(\mathbf{z})\big]. \qquad (27)$$

When $\mathcal{B}_M$ is reasonably large, then $L^{\text{WiSE-ALE}}(\phi, \theta; \mathcal{B}_M)$ becomes an good approximation of $L^{\text{WiSE-ALE}}(\phi, \theta; \mathcal{D}_N)$ through

$$L^{\text{WiSE-ALE}}(\phi, \theta; \mathcal{D}_N) \approx \frac{N}{M} L^{\text{WiSE-ALE}}(\phi, \theta; \mathcal{B}_M). \qquad (28)$$

Given the expressions for the objective functions derived in Section 3.3, we can compute the gradient for an approximation to the lower bound of a mini-batch $\mathcal{B}_M$ and apply stochastic gradient ascent algorithm to iteratively optimize the parameters $\phi$ and $\theta$. We can thus apply our WiSE-ALE algorithm efficiently to a mini-batch and learn a meaningful internal representation of the entire dataset. Algorithmically, WiSE-ALE is similar to AEVB, save for an alternate objective function as per Section 3.3.3. The procedural details of the algorithm are presented in Appendix C.

## 4   RELATED WORK

Bengio et al. (2013) proposes that a learned representation of data should exhibit some generally preferable features, such as smoothness, sparsity and simplicity. However, these attributes are not tailored to any specific downstream tasks. Bayesian decision making (see e.g. Lacoste-Julien et al.

(2011); Cobb et al. (2018)) requires consideration of a target task and proposes that any involved latent distribution approximations should be optimised for the performance over the target task, as well as conforming to the more general properties. The AEVB algorithm (Kingma & Welling, 2013) learns the latent posterior distribution under a reconstruction task, while simultaneously satisfying a prior constraint, which ensures the representation is smooth and compact. However, the prior constraint of the AEVB algorithm imposes significant influence on the solution space (as discussed in Section 3.4), and leads to a sacrifice of reconstruction quality. Our WiSE-ALE algorithm, however, prioritises the reconstruction task yet still enables globally desirable properties.

WiSE-ALE is not the only algorithm that considers an alternate prior form to mitigate its impact on the reconstruction quality. The Gaussian Mixture VAE (Dilokthanakul et al., 2016) uses a Gaussian mixture model to parameterise $p(\mathbf{z})$, encouraging more flexible sample posteriors. The Adversarial Auto-Encoder (Makhzani et al., 2016) matches the aggregate posterior over the latent variables with a prior distribution through adversarial training. The WAE (Tolstikhin et al., 2017) minimises a penalised form of the Wasserstein distance between the aggregate posterior distribution and the prior, claiming a generalisation of the AAE algorithm under the theory of optimal transport (Villani, 2008). More recently, the Sinkhorn Auto-Encoder (Patrini et al., 2018) builds a formal analysis of auto-encoders using an optimal transport based prior and uses the Sinkhorn algorithm as an alternative to estimate the Wasserstein distance in WAE.

Our work differs from these in two main aspects. Firstly, our objective function can be evaluated analytically, leading to an efficient optimization process. In many of the above work, the optimization involves adversarial training and some hyper-parameter tuning, which leading to less efficient learning or even no convergence. Secondly, our WiSE-ALE algorithm naturally finds a balance between good reconstruction quality and preferred latent representation properties, such as smoothness and compactness, as shown in Fig. 1(c). In contrast, some other work sacrifice the properties of smoothness and compactness severely for improved reconstruction quality, as shown in Fig. 1(b). Many works (Bloesch et al., 2018; Clark et al., 2018) have indicated that those properties of the learnt latent representation are essential for tasks that require optimisation over the latent space.

## 5 EXPERIMENTS

We evaluate our WiSE-ALE algorithm in comparison with AEVB, $\beta$-VAE and WAE on the following 3 datasets. The implementation details for all experiments are given in Appendix E.

1. **Sine Wave**. We generated 200,000 sine waves with small random noise: $x(t) = A\sin(2\pi ft + \varphi) + \epsilon$, each containing 256 samples, with independently sampled frequency $f \sim \text{Unif}(0, 20\text{Hz})$, phase angle $\varphi \sim \text{Unif}(0, 2\pi)$ and amplitude $A \sim \text{Unif}(0, 2)$.

2. **MNIST** (LeCun, 1998). 70,000 $28 \times 28$ binary images that contain hand-written digits.

3. **CelebA** (Liu et al., 2015). 202,599 RGB images of aligned celebrity faces of $218 \times 178$ are cropped to square images of $178 \times 178$ and resized to $64 \times 64$.

### 5.1 RECONSTRUCTION QUALITY

Throughout all experiments, our method has shown consistently superior reconstruction quality compared to AEVB, $\beta$-VAE and WAE. Fig. 5 offers a graphical comparison across the reconstructed samples given by different methods for the sine wave and CelebA datasets. For the sine wave dataset, our WiSE-ALE algorithms achieves almost perfect reconstruction, whereas AEVB and $\beta$-VAE often struggle with low-frequency signals and have difficulty predicting the amplitude correctly. For the CelebA dataset, our WiSE-ALE manages to predict much sharper human faces, whereas the AEVB predictions are often blurry and personal characteristics are often ignored. WAE reaches a similar level of reconstruction quality to ours in some images, but it sometimes struggles with discovering the right elevation and azimuth angles, as shown in the second to the right column in Fig. 5b.

### 5.2 PROPERTIES OF THE LEARNT REPRESENTATION SPACE

We understand that a good latent representation should not only reconstruct well, but also preserve some preferable qualities, such as smoothness, compactness and possibly meaningful interpretation of

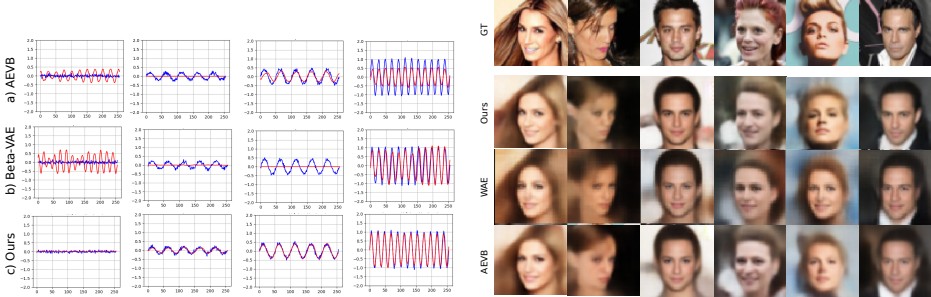

(a) Reconstructed sine waves given by AEVB, $\beta$-VAE and our WiSE-ALE.

(b) Reconstructed celebrity faces given by AEVB, WAE and our WiSE-ALE (CelebA dataset).

Figure 5: Qualitative comparison of the reconstruction quality between our WiSE-ALE and other methods.

the original data. Fig. 1 indicates that our WiSE-ALE automatically learns a latent representation that finds a good tradeoff between minimizing the information loss and maintaining a smooth and compact aggregate posterior distribution. Furthermore, as shown in Fig. 6, we compare the ELBO values given by AEVB, $\beta$-VAE and our WiSE-ALE over training for the Sine dataset. Our WiSE-ALE manages to report the highest ELBO with a significantly lower reconstruction error and a fairly good performance in the KL divergence loss. This indicates that our WiSE-ALE is able to learn an overall good quality representation that is closest to the true latent distribution which gives rise to the data observation.

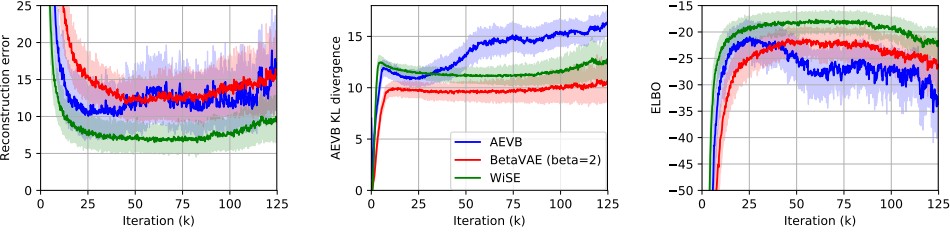

Figure 6: Comparison of training reconstruction error, AEVB KL divergence loss and ELBO given by AEVB, $\beta$-VAE and our WiSE-ALE methods on the sine wave dataset over 50 epochs (batch size = 64).

## 6    CONCLUSION AND FUTURE WORK

In this paper, we propose a new latent variable model where a global latent variable is used to generate the entire dataset. We then derive a variational lower bound to the data log likelihood, which allows us to impose a prior constraint on the *bulk statistics* of the aggregate posterior distribution for the entire dataset. Using an analytic approximation to this lower bound as our learning objective, we propose WiSE-ALE algorithm. We have demonstrated its ability to achieve excellent reconstruction quality, as well as forming a smooth, compact and meaningful latent representation. In the future, we would like to understand the properties of the latent embeddings learnt through our method and apply it for suitable applications.

### ACKNOWLEDGMENTS

The authors would like to thank EPSRC Centre for Doctoral Training in Autonomous Intelligent Machines and Systems EP/L015897/1, University of Oxford, ABB Corporate Research, Switzerland and Swiss National Science Foundation (project 407540_167266) for supporting and funding the research.

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

# APPENDIX FOR WISE-ALE

In this appendix, we omit the trainable parameters $\phi$ and $\theta$ in the expressions of distributions for simplicity. For example, $q(\mathbf{z}|\boldsymbol{x})$ is equivalent to $q_\phi(\mathbf{z}|\boldsymbol{x})$ and $p(\mathbf{x}|\boldsymbol{z})$ represents $p_\theta(\mathbf{x}|\boldsymbol{z})$.

## A APPROXIMATION OF THE RECONSTRUCTION TERM

Here we demonstration that the reconstruction term $\mathbb{E}_{q(\mathbf{z}|\mathcal{D}_N)}\big[\log p(\mathcal{D}_N|\boldsymbol{z})\big]$ in our lower bound can be computed with individual sample likelihood $\log p(\boldsymbol{x}^{(i)}|\boldsymbol{z})$ and how our reconstruction error term becomes the same as the reconstruction term in the AEVB objective.

Firstly, we can substitute

$$q(\mathbf{z}|\mathcal{D}_N) = \frac{1}{N}\sum_{i=1}^{N} q(\mathbf{z}|\boldsymbol{x}^{(i)}) \tag{1}$$

into the reconstruction term $\mathbb{E}_{q(\mathbf{z}|\mathcal{D}_N)}\big[\log p(\mathcal{D}_N|\boldsymbol{z})\big]$, i.e.

$$\begin{aligned}
\mathbb{E}_{q(\mathbf{z}|\mathcal{D}_N)}\big[\log p(\mathcal{D}_N|\boldsymbol{z})\big] &= \int q(\mathbf{z}|\mathcal{D}_N)\big[\log p(\mathcal{D}_N|\boldsymbol{z})\big]\mathrm{d}\mathbf{z} \\
&= \int \frac{1}{N}\sum_{i=1}^{N} q(\mathbf{z}|\boldsymbol{x}^{(i)})\big[\log p(\mathcal{D}_N|\boldsymbol{z})\big]\mathrm{d}\mathbf{z} \\
&= \frac{1}{N}\sum_{i=1}^{N}\int q(\mathbf{z}|\boldsymbol{x}^{(i)})\big[\log p(\mathcal{D}_N|\boldsymbol{z})\big]\mathrm{d}\mathbf{z} \\
&= \frac{1}{N}\sum_{i=1}^{N}\mathbb{E}_{q(\mathbf{z}|\boldsymbol{x}^{(i)})}\big[\log p(\mathcal{D}_N|\boldsymbol{z})\big].
\end{aligned}$$

Now we can decompose the the marginal likelihood of the entire dataset as a product of individual samples, due to the conditional independence, i.e.

$$\log p(\mathcal{D}_N|\boldsymbol{z}) = \log\prod_{j=1}^{N} p(\boldsymbol{x}^{(j)}|\boldsymbol{z}) = \sum_{j=1}^{N}\log p(\boldsymbol{x}^{(j)}|\boldsymbol{z}).$$

Substituting this into the reconstruction term, we have:

$$\mathbb{E}_{q(\mathbf{z}|\mathcal{D}_N)}[\log p(\mathcal{D}_N|\boldsymbol{z})] = \frac{1}{N}\sum_{i=1}^{N}\mathbb{E}_{q(\mathbf{z}|\boldsymbol{x}^{(i)})}\left[\sum_{j=1}^{N}\log p(\boldsymbol{x}^{(j)}|\boldsymbol{z})\right].$$

To evaluate the reconstruction term in our lower bound, we need to do the following: 1) draw a sample $\boldsymbol{x}^{(i)}$ from the dataset $\mathcal{D}_N$; 2) evaluate the latent code distribution $q(\mathbf{z}|\boldsymbol{x}^{(i)})$ through the encoder function $q(\cdot|\boldsymbol{x}^{(i)})$; 3) draw samples of $\mathbf{z}$ according to $q(\mathbf{z}|\boldsymbol{x}^{(i)})$; 4) reconstruct input samples using the sampled latent codes $\boldsymbol{z}^{(l)}$; 5) compute the reconstruction error w.r.t to every single input sample and sum this error.

We can simplify the above evaluation. Firstly, the sampling process in Step 3 can be replaced to a sampling process at the input using the reparameterisation trick. Besides, the sum of reconstruction errors w.r.t. all the input samples can be further simplified. To do this, we need to re-arrange the above expression as

$$\mathbb{E}_{q(\mathbf{z}|\mathcal{D}_N)}[\log p(\mathcal{D}_N|\boldsymbol{z})] = \sum_{i=1}^{N}\mathbb{E}_{q(\mathbf{z}|\boldsymbol{x}^{(i)})}\left[\frac{1}{N}\sum_{j=1}^{N}\log p(\boldsymbol{x}^{(j)}|\boldsymbol{z})\right]$$

and apply Jensen inequality for the special case of log, i.e.

$$\log\left(\frac{1}{N}\sum_{i=1}^{N}a_i\right) \geq \frac{1}{N}\sum_{i=1}^{N}\log(a_i)$$

to the terms inside the expectation. As a result, we have obtain an upper bound of the reconstruction error term as

$$\mathbb{E}_{q(\mathbf{z}|\mathcal{D}_N)}[\log p(\mathcal{D}_N|\mathbf{z})] \leq \sum_{i=1}^{N}\mathbb{E}_{q(\mathbf{z}|\mathbf{x}^{(i)})}\left[\log\left(\frac{1}{N}\sum_{j=1}^{N}p(\mathbf{x}^{(j)}|\mathbf{z})\right)\right].$$

This upper bound can be evaluated more efficiently with the assumption that the likelihood $p(\mathbf{x}^{(j)}|\mathbf{z})$ representing the probability of a reconstructed sample from a latent code $\mathbf{z}$ imitating the sample $\mathbf{x}^{(j)}$ will only be non-zero if $\mathbf{z}$ is sampled from the embedding prediction distribution with the same sample $\mathbf{x}^{(j)}$ at the encoder input. With this assumption, $N-1$ posterior distributions in the inner summation will be dropped as zeros and the only non-zero term is $p(\mathbf{x}^{(i)}|\mathbf{z})$. Therefore, the upper bound becomes

$$\mathbb{E}_{q(\mathbf{z}|\mathcal{D}_N)}[\log p(\mathcal{D}_N|\mathbf{z})] \leq \sum_{i=1}^{N}\mathbb{E}_{q(\mathbf{z}|\mathbf{x}^{(i)})}\left[\log\left(\frac{1}{N}p(\mathbf{x}^{(i)}|\mathbf{z})\right)\right]$$

$$= \sum_{i=1}^{N}\mathbb{E}_{q(\mathbf{z}|\mathbf{x}^{(i)})}\left[\log p(\mathbf{x}^{(i)}|\mathbf{z})\right] - N\log N$$

$$\approx \sum_{i=1}^{N}\mathbb{E}_{q(\mathbf{z}|\mathbf{x}^{(i)})}\left[\log p(\mathbf{x}^{(i)}|\mathbf{z})\right].$$

The constant can be omitted, because it will not affect the gradient updates of the parameters.

## B   AN UPPER BOUND APPROXIMATION OF THE KL TERM

$$D_{\text{KL}}\big[q(\mathbf{z}|\mathcal{D}_N)\|p(\mathbf{z})\big] = \int q(\mathbf{z}|\mathcal{D}_N)\Big(\log q(\mathbf{z}|\mathcal{D}_N) - \log p(z)\Big)\mathrm{d}\mathbf{z}$$

$$= \int \frac{1}{N}\sum_{i=1}^{N}q(\mathbf{z}|\mathbf{x}^{(i)})\Big(\log q(\mathbf{z}|\mathcal{D}_N) - \log p(\mathbf{z})\Big)\mathrm{d}\mathbf{z}$$

$$= \frac{1}{N}\sum_{i=1}^{N}\int q(\mathbf{z}|\mathbf{x}^{(i)})\Big(\log q(\mathbf{z}|\mathcal{D}_N) - \log p(\mathbf{z})\Big)\mathrm{d}\mathbf{z}$$

$$= \frac{1}{N}\sum_{i=1}^{N}\Big(\mathbb{E}_{q(\mathbf{z}|\mathbf{x}^{(i)})}\big[\log q(\mathbf{z}|\mathcal{D}_N)\big] - \mathbb{E}_{q(\mathbf{z}|\mathbf{x}^{(i)})}\big[\log p(\mathbf{z})\big]\Big)$$

Applying Jensen inequality, i.e.

$$\mathbb{E}_x\big[\log f(x)\big] \leq \log\mathbb{E}_x\big[f(x)\big], \tag{2}$$

to the first term of above equation, we have

$$D_{\text{KL}}\big[q(\mathbf{z}|\mathcal{D}_N)\|p(\mathbf{z})\big] \leq \frac{1}{N}\sum_{i=1}^{N}\Big(\log\mathbb{E}_{q(\mathbf{z}|\mathbf{x}^{(i)})}\big[q(\mathbf{z}|\mathcal{D}_N)\big]\Big) - \frac{1}{N}\sum_{i=1}^{N}\Big(\mathbb{E}_{q(\mathbf{z}|\mathbf{x}^{(i)})}\big[\log p(\mathbf{z})\big]\Big).$$

We will look at the two summation individually. The expectation w.r.t. the aggregate posterior can be expanded as

$$\mathbb{E}_{q(\mathbf{z}|\mathbf{x}^{(i)})}\big[q(\mathbf{z}|\mathcal{D}_N)\big] = \int q(\mathbf{z}|\mathbf{x}^{(i)})\frac{1}{N}\sum_{j=1}^{N}q(\mathbf{z}|\mathbf{x}^{(j)})\,\mathrm{d}\mathbf{z}$$

$$= \frac{1}{N}\sum_{j=1}^{N}\int q(\mathbf{z}|\mathbf{x}^{(i)})\,q(\mathbf{z}|\mathbf{x}^{(j)})\,\mathrm{d}\mathbf{z}.$$

We assume the posterior distribution of the latent code $\mathbf{z}$ given a specific input sample $\boldsymbol{x}^{(i)}$ is a diagonal Gaussian, i.e.

$$q(\mathbf{z} \,|\, \boldsymbol{x}^{(i)}) = \mathcal{N}\Big(\mathbf{z} \,|\, \mu^{(i)}, (\sigma^{(i)})^2\Big) = \prod_{k=1}^{d_z} \mathcal{N}\Big(\mathbf{z}_k \,|\, \mu_k^{(i)}, (\sigma_k^{(i)})^2\Big). \tag{3}$$

Similarly,

$$q(\mathbf{z} \,|\, \boldsymbol{x}^{(j)}) = \mathcal{N}\Big(\mathbf{z} \,|\, \mu^{(j)}, (\sigma^{(j)})^2\Big) = \prod_{k=1}^{d_z} \mathcal{N}\Big(\mathbf{z}_k \,|\, \mu_k^{(j)}, (\sigma_k^{(j)})^2\Big).$$

Therefore,

$$
\begin{aligned}
\mathbb{E}_{q(\mathbf{z}|\boldsymbol{x}^{(i)})}\big[q(\mathbf{z}|\mathcal{D}_N)\big] &= \frac{1}{N}\sum_{j=1}^{N}\int \prod_{k=1}^{d_z}\mathcal{N}\Big(\mathbf{z}_k \,|\, \mu_k^{(i)},(\sigma_k^{(i)})^2\Big) \prod_{k=1}^{d_z}\mathcal{N}\Big(\mathbf{z}_k \,|\, \mu_k^{(j)},(\sigma_k^{(j)})^2\Big) \prod_{k=1}^{d_z}\mathrm{d}z_k \\
&= \frac{1}{N}\sum_{j=1}^{N}\prod_{k=1}^{d_z}\int \mathcal{N}\Big(\mathbf{z}_k \,|\, \mu_k^{(i)},(\sigma_k^{(i)})^2\Big) \mathcal{N}\Big(\mathbf{z}_k \,|\, \mu_k^{(j)},(\sigma_k^{(j)})^2\Big) \mathrm{d}\mathbf{z}_k.
\end{aligned}
$$

Substituting the exponential form for Gaussian distribution, i.e.

$$\mathcal{N}\Big(\mathbf{z}_k \,|\, \mu_k^{(i)},(\sigma_k^{(i)})^2\Big) = \frac{1}{\sqrt{2\pi(\sigma_k^{(i)})^2}}\exp\left(-\frac{(z_k-\mu_k^{(i)})^2}{2(\sigma_k^{(i)})^2}\right), \tag{4}$$

to the above equation, we have

$$
\begin{aligned}
\mathbb{E}_{q(\mathbf{z}|\boldsymbol{x}^{(i)})}\big[q(\mathbf{z}|\mathcal{D}_N)\big] &= \frac{1}{N}\sum_{j=1}^{N}\prod_{k=1}^{d_z}\int \frac{1}{2\pi\sigma_k^{(i)}\sigma_k^{(j)}}\exp\left(-\frac{(z_k-\mu_k^{(i)})^2}{2(\sigma_k^{(i)})^2}-\frac{(z_k-\mu_k^{(j)})^2}{2(\sigma_k^{(j)})^2}\right)\mathrm{d}z_k \\
&= \frac{1}{N}\sum_{j=1}^{N}\prod_{k=1}^{d_z}\frac{1}{2\pi\sigma_k^{(i)}\sigma_k^{(j)}}\int \exp\left(-\frac{(z_k-\mu_k^{(i)})^2}{2(\sigma_k^{(i)})^2}-\frac{(z_k-\mu_k^{(j)})^2}{2(\sigma_k^{(j)})^2}\right)\mathrm{d}z_k.
\end{aligned}
$$

The exponent of the above equation can be simplified to

$$
\begin{aligned}
&-\frac{(z_k-\mu_k^{(i)})^2}{2(\sigma_k^{(i)})^2}-\frac{(z_k-\mu_k^{(j)})^2}{2(\sigma_k^{(j)})^2} \\
&= -\frac{1}{2}\left(\frac{1}{(\sigma_k^{(i)})^2}+\frac{1}{(\sigma_k^{(j)})^2}\right)z_k^2 + \left(\frac{\mu_k^{(i)}}{(\sigma_k^{(i)})^2}+\frac{\mu_k^{(j)}}{(\sigma_k^{(j)})^2}\right)z_k - \frac{1}{2}\left(\frac{(\mu_k^{(i)})^2}{(\sigma_k^{(i)})^2}+\frac{(\mu_k^{(i)})^2}{(\sigma_k^{(i)})^2}\right).
\end{aligned}
$$

Using the following properties, i.e.

$$\int_{-\infty}^{\infty}\exp\big(-ax^2+bx\big)\,\mathrm{d}x = \sqrt{\frac{\pi}{a}}\exp\left(\frac{b^2}{4a}\right) \quad (a\geq 0), \tag{5}$$

we can evaluate the integral needed for $\mathbb{E}_{q(z|x^{(i)})}\big[q(z|\mathcal{D}_N)\big]$ as

$$
\begin{aligned}
&\frac{1}{2\pi\sigma_k^{(i)}\sigma_k^{(j)}}\int_{z_k}\exp\left(-\frac{(z_k-\mu_k^{(i)})^2}{2(\sigma_k^{(i)})^2}-\frac{(z_k-\mu_k^{(j)})^2}{2(\sigma_k^{(j)})^2}\right)\mathrm{d}z_k \\
&\qquad = \frac{1}{\sqrt{2\pi((\sigma_k^{(i)})^2+(\sigma_k^{(j)})^2)}}\exp\left(-\frac{1}{2}\frac{(\mu_k^{(i)}-\mu_k^{(j)})^2}{(\sigma_k^{(i)})^2+(\sigma_k^{(j)})^2}\right).
\end{aligned}
$$

Therefore, we have obtained the expression for the first term in our upper bound, i.e.

$$\frac{1}{N}\sum_{i=1}^{N}\Big(\log \mathbb{E}_{q(\mathbf{z}|\boldsymbol{x}^{(i)})}\big[q(\mathbf{z}|\mathcal{D}_N)\big]\Big) \tag{6}$$

$$= \frac{1}{N}\sum_{i=1}^{N}\log\left(\frac{1}{N}\sum_{j=1}^{N}\prod_{k=1}^{d_z}\frac{1}{\sqrt{2\pi((\sigma_k^{(i)})^2+(\sigma_k^{(j)})^2)}}\exp\left(-\frac{1}{2}\frac{(\mu_k^{(i)}-\mu_k^{(j)})^2}{(\sigma_k^{(i)})^2+(\sigma_k^{(j)})^2}\right)\right). \tag{7}$$

To find out the expression for the second term $\frac{1}{N}\sum_{i=1}^{N}\left(\mathbb{E}_{q(\mathbf{z}|\boldsymbol{x}^{(i)})}\big[\log p(\mathbf{z})\big]\right)$, we first examine the prior distribution $p(\mathbf{z})$ which is chosen to be a zero-mean unit-variance Gaussian across all latent code dimensions, i.e.

$$p(\mathbf{z}) \;=\; \mathcal{N}\big(\mathbf{z}\,|\,0,\,\mathcal{I}\big) \;=\; \prod_{k=1}^{d_z}\mathcal{N}\big(\mathbf{z}_k\,|\,0,1\big). \tag{8}$$

Therefore,

$$\log p(\mathbf{z}) = \sum_{k=1}^{d_z}\log\mathcal{N}\big(\mathbf{z}_k\,|\,0,1\big) = -\frac{1}{2}\sum_{k=1}^{d_z}\Big(\log\big(2\pi\big)+z_k^2\Big) \tag{9}$$

Substituting this expression for $\log p(\mathbf{z})$ into $\frac{1}{N}\sum_{i=1}^{N}\left(\mathbb{E}_{q(\mathbf{z}|\boldsymbol{x}^{(i)})}\big[\log p(\mathbf{z})\big]\right)$ and examining the expectation term for now, we have

$$
\begin{aligned}
\mathbb{E}_{q(\mathbf{z}|\boldsymbol{x}^{(i)})}\big[\log p(\mathbf{z})\big]
&= \sum_{k=1}^{d_z}\mathbb{E}_{q(\mathbf{z}|\boldsymbol{x}^{(i)})}\big[\log p(\mathbf{z}_k)\big] \\
&= \sum_{k=1}^{d_z}\mathbb{E}_{q(\mathbf{z}_k|\boldsymbol{x}^{(i)})q(\mathbf{z}_{\backslash k}|\boldsymbol{x}^{(i)})}\big[\log p(\mathbf{z}_k)\big] \\
&= \sum_{k=1}^{d_z}\mathbb{E}_{q(\mathbf{z}_k|\boldsymbol{x}^{(i)})}\big[\log p(\mathbf{z}_k)\big] \\
&= \sum_{k=1}^{d_z}\int q(\mathbf{z}_k|\boldsymbol{x}^{(i)})\log p(\mathbf{z}_k)\,\mathrm{d}z_k \\
&= -\frac{1}{2}\sum_{k=1}^{d_z}\int q(\mathbf{z}_k|\boldsymbol{x}^{(i)})\Big(\log\big(2\pi\big)+z_k^2\Big)\,\mathrm{d}z_k \\
&= -\frac{1}{2}\sum_{k=1}^{d_z}\left(\log\big(2\pi\big)\int q(\mathbf{z}_k|\boldsymbol{x}^{(i)})\,\mathrm{d}z_k \;+\; \int q(\mathbf{z}_k|\boldsymbol{x}^{(i)})z_k^2\,\mathrm{d}z_k\right).
\end{aligned}
$$

The first integral $\int q(\mathbf{z}_k|\boldsymbol{x}^{(i)})\,\mathrm{d}z_k = 1$. To evaluate the second integral, we substitute Equation (4) and use the following properties, i.e.

$$\int_{-\infty}^{\infty}\exp\big(-ax^2\big)\,\mathrm{d}x \;=\; \frac{1}{2}\sqrt{\frac{\pi}{a}},\quad a\geq 0 \tag{10}$$

$$\int_{-\infty}^{\infty}x\exp\big(-a(x-b)^2\big)\,\mathrm{d}x \;=\; b\sqrt{\frac{\pi}{a}},\quad \mathrm{Re}(a)\geq 0 \tag{11}$$

$$\int_{-\infty}^{\infty}x^2\exp\big(-ax^2\big)\,\mathrm{d}x \;=\; \frac{1}{2}\sqrt{\frac{\pi}{a^3}},\quad a\geq 0. \tag{12}$$

As a result, we have

$$\int_{z_k} q(z_k|\boldsymbol{x}^{(i)})z_k^2\,\mathrm{d}z_k \;=\; \big(\sigma_k^{(i)}\big)^2 + \big(\mu_k^{(i)}\big)^2.$$

Therefore,

$$\mathbb{E}_{q(\mathbf{z}|\boldsymbol{x}^{(i)})}\big[\log p(\mathbf{z})\big] \;=\; -\frac{1}{2}\sum_{k=1}^{d_z}\left(\big(\sigma_k^{(i)}\big)^2 + \big(\mu_k^{(i)}\big)^2 + \log 2\pi\right) \tag{13}$$

$$\frac{1}{N}\sum_{i=1}^{N}\mathbb{E}_{q(\mathbf{z}|\boldsymbol{x}^{(i)})}\big[\log p(\mathbf{z})\big] \;=\; -\frac{1}{2N}\sum_{i=1}^{N}\sum_{k=1}^{d_z}\left(\big(\sigma_k^{(i)}\big)^2 + \big(\mu_k^{(i)}\big)^2 + \log 2\pi\right). \tag{14}$$

Combining the first term defined in Equation (6) and the second term defined in Equation (13), we have obtained the expression for the overall upper bound as

$$D_{\mathrm{KL}}\big[\, q(\mathbf{z}|\mathcal{D}_N)\|p(\mathbf{z})\big]$$

$$\leq \frac{1}{N}\sum_{i=1}^{N}\log\left(\frac{1}{N}\sum_{j=1}^{N}\prod_{k=1}^{d_z}\frac{1}{\sqrt{2\pi\big((\sigma_k^{(i)})^2+(\sigma_k^{(j)})^2\big)}}\exp\left(-\frac{1}{2}\frac{\big(\mu_k^{(i)}-\mu_k^{(j)}\big)^2}{(\sigma_k^{(i)})^2+(\sigma_k^{(j)})^2}\right)\right)$$

$$+\frac{1}{2N}\sum_{i=1}^{N}\sum_{k=1}^{d_z}\left((\sigma_k^{(i)})^2+(\mu_k^{(i)})^2+\log 2\pi\right).$$

## C  WiSE Algorithm

---

**Algorithm 1** WiSE algorithm. Either $L_{\mathrm{approx}}^{\mathrm{WiSE}}(\phi,\theta;\mathcal{B}_M)$ defined in Eq. 19 in Section 3.5 can be used as the learning objective function.

---

$\phi,\theta \leftarrow$ Initialize parameters
**repeat**
    $\mathcal{B}_M \leftarrow$ Draw $M$ random samples from $\mathcal{D}_N$
    $\epsilon \leftarrow \mathcal{N}(0,\boldsymbol{I})$ Apply reparameterisation trick so that $\boldsymbol{z}\sim q_\phi(\mathbf{z}|\boldsymbol{x}^{(i)})$ becomes $\boldsymbol{z}=\mu^{(i)}+\epsilon\odot\sigma^{(i)}$

    $g \leftarrow \bigtriangledown_{\phi,\theta} L_{\mathrm{approx}}^{\mathrm{WiSE}}(\phi,\theta;\mathcal{B}_M)$ Compute the gradient
    $\phi,\theta \leftarrow$ Update parameters using $g$ according to AdamOptimizer
**until** convergence of the objective function or end of iterations
**return** $\phi,\theta$

---

## D  Experiment Details

We carry out experiments on four datasets (Sine wave, MNIST, Teapot and CelebA) to examine different properties of the latent representation learnt from the proposed WiSE algorithm. Specifically, we compare with $\beta$-VAE on the smoothness and disentanglement of the learnt representation and compare with WAE and AEVB on the reconstruction quality. In addition, by learning a 2D embedding of the MNIST dataset, we are able to visualise the latent embedding distributions learnt from AEVB, $\beta$-VAE, WAE and our WiSE and compare the compactness and smoothness of the learnt latent space across these methods. Here we give the implementation details for each dataset.

### D.1  Sine Wave

We aim to learn a latent representation in $\mathcal{R}^4$ for a one second long sine wave with sampling rate of 256Hz. The network architecture for the Sine wave dataset is shown below. $\boldsymbol{x}$ is an input sample, $\mu$ and $\sigma$ are the latent code mean and latent code standard deviation to define the embedding distribution $q(\mathbf{z}|\boldsymbol{x})$ and $\hat{\boldsymbol{x}}$ is the reconstructed input sample. $\epsilon$ is an auxiliary variable drawn from unit Gaussian at the input of the encoder network so that an estimate of a sample from the embedding distribution $q(\mathbf{z}|\boldsymbol{x})$ can be computed. $\mathrm{Conv}_k^{m\times n}$ denotes a convolution operation with $k$ filters each of size $m\times n$. $\mathrm{TransposedConv}_k^{m\times n}$ (stride$\,=$(a,b)) denotes a stride of $a$ and $b$ for the sliding window and $k$ filters each of size $m\times n$. $\mathrm{FC}_k$ denotes a fully connected layer with output in $\mathcal{R}^k$. $\mathrm{Reshape}_a^b$ denotes reshaping an variable from dimension $a$ to dimension $b$. ReLU denotes rectified linear units.

Encoder network:

$$\boldsymbol{x} \in \mathcal{R}^{256 \times 1} \to \mathrm{Conv}_{16}^{16 \times 1} \, (\mathrm{stride} = 2) \to \mathrm{ReLU}$$
$$\to \mathrm{Conv}_{16}^{16 \times 1} \, (\mathrm{stride} = 2) \to \mathrm{ReLU}$$
$$\to \mathrm{Conv}_{32}^{16 \times 1} \, (\mathrm{stride} = 2) \to \mathrm{ReLU}$$
$$\to \mathrm{Conv}_{32}^{16 \times 1} \, (\mathrm{stride} = 2) \to \mathrm{ReLU}$$
$$\to \mathrm{Conv}_{64}^{8 \times 1} \, (\mathrm{stride} = 2) \to \mathrm{ReLU} \to \mathrm{FC}_{64} \to \mathrm{FC}_4 \Rightarrow \ \mu \in \mathcal{R}^4$$
$$\searrow$$
$$\mathrm{FC}_4 \to \mathrm{ReLU} \Rightarrow \ \sigma \in \mathcal{R}^4$$

Decoder network:

$$\boldsymbol{z} = \mu + \epsilon \odot \sigma \to \mathrm{FC}_{16} \to \mathrm{ReLU} \to \mathrm{Reshape}_{16}^{1 \times 1 \times 16}$$
$$\to \mathrm{Conv}_{128}^{1 \times 1} \to \mathrm{ReLU}$$
$$\to \mathrm{TransposedConv}_{64}^{8 \times 1} \, (\mathrm{stride} = (4,1)) \to \mathrm{ReLU}$$
$$\to \mathrm{TransposedConv}_{32}^{16 \times 1} \, (\mathrm{stride} = (4,1)) \to \mathrm{ReLU}$$
$$\to \mathrm{TransposedConv}_{16}^{16 \times 1} \, (\mathrm{stride} = (4,1)) \to \mathrm{ReLU}$$
$$\to \mathrm{TransposedConv}_{1}^{16 \times 1} \, (\mathrm{stride} = (4,1)) \to \mathrm{ReLU} \to \mathrm{Reshape}_{256 \times 1 \times 1}^{256 \times 1} \Rightarrow \ \hat{\boldsymbol{x}} \in \mathcal{R}^{256 \times 1}$$

We use the following hyper-parameters to train the network:

| Batch size | Number of epochs | Optimizer | Learning rate | Padding |
|---|---|---|---|---|
| 64 | 50 | Adam | $5 \times 10^{-4}$ | SAME |

## D.2  MNIST

We aim to learn a 2D embedding of the MNIST dataset. The network architecture is shown below.

Encoder network:

$$\boldsymbol{x} \in \mathcal{R}^{28 \times 28 \times 1} \to \mathrm{Conv}_{16}^{4 \times 4} \, (\mathrm{stride} = 2) \to \mathrm{ReLU}$$
$$\to \mathrm{Conv}_{32}^{4 \times 4} \, (\mathrm{stride} = 2) \to \mathrm{ReLU}$$
$$\to \mathrm{Conv}_{64}^{4 \times 4} \, (\mathrm{stride} = 2) \to \mathrm{ReLU} \to \mathrm{FC}_{32} \to \mathrm{FC}_2 \Rightarrow \ \mu \in \mathcal{R}^2$$
$$\searrow$$
$$\mathrm{FC}_2 \to \mathrm{ReLU} \Rightarrow \ \sigma \in \mathcal{R}^2$$

Decoder network:

$$\boldsymbol{z} = \mu + \epsilon \odot \sigma \to \mathrm{FC}_{16} \to \mathrm{ReLU}$$
$$\to \mathrm{FC}_{128} \to \mathrm{ReLU}$$
$$\to \mathrm{FC}_{784} \to \mathrm{Sigmoid} \to \mathrm{Reshape}_{784}^{28 \times 28 \times 1} \Rightarrow \ \hat{\boldsymbol{x}} \in \mathcal{R}^{28 \times 28 \times 1}$$

We use the following hyper-parameters to train the network:

| Batch size | Number of epochs | Optimizer | Learning rate | Padding |
|---|---|---|---|---|
| 64 | 30 | Adam | $1 \times 10^{-3}$ | SAME |

## D.3  CELEBA

We implement our WiSE and AEVB on the same encoder and decoder network used in WAE in order to compare the reconstruction quality of our method with AEVB and WAE. The network architecture and training parameters are stated below.

Encoder network:

$$\boldsymbol{x} \in \mathcal{R}^{64 \times 64 \times 3} \to \mathrm{Conv}_{128}^{5 \times 5} \; (\mathrm{stride} = 2) \to \mathrm{ReLU}$$
$$\to \mathrm{Conv}_{256}^{5 \times 5} \; (\mathrm{stride} = 2) \to \mathrm{ReLU}$$
$$\to \mathrm{Conv}_{512}^{5 \times 5} \; (\mathrm{stride} = 2) \to \mathrm{ReLU}$$
$$\to \mathrm{Conv}_{1024}^{5 \times 5} \; (\mathrm{stride} = 2) \to \mathrm{ReLU} \to \mathrm{FC}_{64} \Rightarrow \; \mu \in \mathcal{R}^{64}$$
$$\searrow$$
$$\mathrm{FC}_{64} \Rightarrow \; \sigma \in \mathcal{R}^{64}$$

Decoder network:

$$\boldsymbol{z} = \mu + \epsilon \odot \sigma \to \mathrm{FC}_{64 \times 1024} \to \mathrm{ReLU} \to \mathrm{Reshape}_{64 \times 1024}^{8 \times 8 \times 1024}$$
$$\to \mathrm{TransposedConv}_{512}^{5 \times 5} \; (\mathrm{stride} = (2,2)) \to \mathrm{BN} \to \mathrm{ReLU}$$
$$\to \mathrm{TransposedConv}_{256}^{5 \times 5} \; (\mathrm{stride} = (2,2)) \to \mathrm{BN} \to \mathrm{ReLU}$$
$$\to \mathrm{TransposedConv}_{128}^{5 \times 5} \; (\mathrm{stride} = (2,2)) \to \mathrm{BN} \to \mathrm{ReLU}$$
$$\to \mathrm{TransposedConv}_{1}^{5 \times 5} \; (\mathrm{stride} = (1,1)) \Rightarrow \; \hat{\boldsymbol{x}} \in \mathcal{R}^{64 \times 64 \times 3}$$

We use the following hyper-parameters to train the network:

| Batch size | Number of epochs | Optimizer | Learning rate | Padding |
|------------|------------------|-----------|---------------|---------|
| 256 | 50 | Adam | $3 \times 10^{-4}$ at the start, $1.5 \times 10^{-4}$ after 30 epochs | SAME |

