# OpenReview forum: "WiSE-ALE: Wide Sample Estimator for Aggregate Latent Embedding"
_ICLR.cc/2019/Workshop/DeepGenStruct — DeepGenStruct 2019_

### Official Review · AnonReviewer1 · 2019-04-16
**A new generative process for VAE**

**Rating:** 4
**Confidence:** 2

**Review:**

This paper presents a new generative model for learning latent embeddings. Compared with the classical generative process, where each observed data point is generated from an individually sampled latent representation from thr prior distribution, the proposed approach assumes a single sampled latent representation to generate the whole set of observed data points.

The authors then propose a learning objective without guarantee of boundness. Compared with the standard ELBO objective, where the variational posterior for each data point is encouraged to matched with the prior distribution, the proposed objective matches the averaged posterior of all samples with the prior, allowing the posterior distribution to have a wider range of acceptable embedding mean and variance, as long as the averaged value over all samples is close to a standard Gaussian (Fig. 3 and 4).

This is a nice submission with reasonable technical contribution and empirical evaluation (against beta-VAE, classical AEVB and WAE) as a workshop paper. It would be great if the authors could further analyze the error introduced in the approximation by Jensen inequality, as outlined in the future work. Meanwhile, a quantitative analysis against WAE (instead of only presenting illustrative examples in Fig. 5(b)) would also be helpful.

---

### Official Review · AnonReviewer2 · 2019-04-16
**solid work**

**Rating:** 4
**Confidence:** 2

**Review:**

This paper derives a variational lower bound to the data log likelihood, which allows us to
impose a prior constraint on the bulk statistics of the aggregate posterior distribution for the entire
dataset. The analysis shows that the proposed method achieves better reconstruction quality, as well as forming a smooth, compact and meaningful latent representation. I would like to accept the paper for the workshop.

---

### Decision · Program_Chairs · 2019-04-19
**Acceptance Decision**

Accept